# Chondrocytes In Vitro Systems Allowing Study of OA

**DOI:** 10.3390/ijms231810308

**Published:** 2022-09-07

**Authors:** Ewa Bednarczyk

**Affiliations:** Faculty of Mechanical and Industrial Engineering, Warsaw University of Technology, Narbutta 85, 02-524 Warsaw, Poland; ewa.bednarczyk@pw.edu.pl

**Keywords:** osteoarthritis, chondrocyte cultures, mechanical loading, tissue engineering, mechanobiology, in-vitro models

## Abstract

Osteoarthritis (OA) is an extremely complex disease, as it combines both biological-chemical and mechanical aspects, and it also involves the entire joint consisting of various types of tissues, including cartilage and bone. This paper describes the methods of conducting cell cultures aimed at searching for the mechanical causes of OA development, therapeutic solutions, and methods of preventing the disease. It presents the systems for the cultivation of cartilage cells depending on the level of their structural complexity, and taking into account the most common solutions aimed at recreating the most important factors contributing to the development of OA, that is mechanical loads. In-vitro systems used in tissue engineering to investigate the phenomena associated with OA were specified depending on the complexity and purposefulness of conducting cell cultures.

## 1. Introduction

Tissue engineering, a field of science investigating cell cultures, tissues, and organs and suggesting alternative solutions that change the approach to treatment, has experienced dynamic development in recent years. Reconstruction of damaged tissues or organs via tissue engineering is a multistep process. Initially, a tissue fragment is collected from the patient’s or another donor’s body. The collected specimen is processed by, for example, releasing the cells from their natural matrix. Then the resulting cells are plated on culture media. At the next stage, the cells are cultured in the medium under specific conditions and they replicate [1]. One of the fields of tissue engineering focus is the research on a civilization disease, osteoarthritis (OA). To improve the therapeutic options for this increasingly prevalent condition, scientists investigate joint building cells and analyze the effects of selected factors on the cells’ behavior. As OA is a complex disease with development affected by numerous factors, in-vitro research is multifaceted and interdisciplinary. Therefore, it is extremely difficult to recreate in a laboratory all the actual conditions the body experiences in the course of the disease. However, with technologies that enable the production of modern culture carriers and innovative techniques for conducting cell cultures, it is possible to make these conditions increasingly similar to those prevailing in-vivo.

This work describes the methods of conducting cell cultures aimed at searching for the causes of OA development, therapeutic solutions, and methods of preventing the disease. It presents the systems for the cultivation of chondrocytes (cartilage cells) depending on the level of their structural complexity, and taking into account the most common solutions aimed at recreating the most important factors contributing to the development of OA, that is mechanical loads, as shown in Figure 1.

## 2. What Is Osteoarthritis

Osteoarthritis, the most common type of arthritis, is one of the most common diseases of civilization in the world [2]. Even though the problem is very common, researchers are yet to fully understand the processes behind the degeneration of joints. It is an extremely complex issue, as it combines both biological-chemical and mechanical aspects, and it also involves the entire joint consisting of various types of tissues, including cartilage and bone [3]. Degenerative changes can occur in any type of joint; however, they most often occur in the knee and hip joints in elderly people, more often in women than in men, as well as in athletes, overweight people, and people with non-physiological posture, for example, with varus or valgus of the lower extremities [4,5].

Degenerative changes in the joints include wearing and loss of articular cartilage, growth of bone spurs, and remodeling of subchondral bone. These changes are accompanied by inflammation. OA can develop in any person but the groups mentioned above are at a greater risk. Osteoarthritis is diagnosed primarily in the elderly, in whom cartilage tissue loss in the joints is observed after many years of functioning. OA causes severe pain and limits normal physical activity of the patients [6]. It is becoming a more and more frequently diagnosed disease, but the very process of its occurrence and development is still not fully understood. The degenerative changes are influenced by aspects related to mechanics, biology and chemistry, as well as the element of randomness depending on the individual characteristics of a given organism. The complexity of the disease makes it an interesting and challenging subject of research [7].

Scientific literature on the development of osteoarthritis has identified the causes and effects of OA, as listed in Table 1 [8,9,10,11,12,13,14,15,16]:

Osteoarthritis develops gradually (Figure 2). Its onset is characterized by a loss of synovial fluid between the articular surfaces covered with hyaline cartilage, which initiates inflammation. The cartilage begins to lose its absorbing properties and its continuity, microcracks appear, but its surface remains smooth. At the next stage, cartilage cells undergo gradual growth (hypertrophy), which leads to their death. This is accompanied by an increasing network of microcracks in the subchondral bone layer. In this phase of the disease, the surface of the cartilage top layer is rough and cracked, which further reduces its shock-absorbing ability. Then, as a result of thinning and delamination, local defects in the top layer of cartilage can be observed. Subsequently, cysts form in the middle layer of the cartilage and osteophytes (bone spurs) appear on the border between the bone and the cartilage. This process is associated with the growth of blood vessels from the subchondral bone into the cartilage [17]. In the final stage of the disease, the middle and basal layers of the cartilage erode. At this point, the cartilage is almost completely worn away and the bone surface is exposed. The bone spurs are now clearly visible and impede the movement of the joint. Moreover, at this stage of the disease, bone remodeling begins [15,18,19].

Bone spur growth is commonly known as osteophytosis. It consists in the growth of bones in the form of so-called osteophytes. During development of degenerative changes in the joint, osteophytes grow from the bone into the cartilage, mostly where the tissues and the joint capsule meet, that is on the cartilage margin (Figure 2). However, central osteophytes [20], formed on articular surfaces, can also be observed.

It is important to remember that degenerative changes in cartilage begin at the cellular level. Chondrocytes, subjected to mechanical stresses causing their hypertrophy and apoptosis, produce an increased amount of biochemical factors, such as vascular endothelial growth factor (VEGF) [15,21,22,23]. VEGF directly induces growth of a new blood vessel network in a process called angiogenesis. As the new blood vessels grow into the cartilage, they provide appropriate conditions for the growth of bone spurs (osteophytes) [21,24,25,26,27].

To study the processes taking place in the cartilage in the course of osteoarthritis, researchers try to recreate in-vivo conditions in their in-vitro tests. The following sections present the in-vitro systems used in tissue engineering to investigate the complex phenomena associated with osteoarthritis. 

## 3. Cell Cultures

Establishing a cell culture for in-vitro studies begins with the collection of selected cells from the body. In a living organism, cells are closely related to the surrounding spatial structures, which control their nutrition, metabolism, morphology, and degradation. It is therefore important to ensure that the growth conditions in the cell culture are as similar to those in-vivo as possible. Those conditions involve a medium of appropriate type and composition, the right temperature, or access to oxygen and carbon dioxide [28,29].

There are two types of cell cultures. The first one is called an adherent culture. It consists of a single layer of cells (monolayer), and is also known as 2D culture. This type of culture is popular, as it is relatively easy to establish and manage. However, it does not sufficiently reproduce the complex nature of reactions occurring in a living organism that is necessary in some experiments. In order to make the processes taking place in the cultures similar to those occurring in-vivo, a second type of culture can be utilized, that is 3D culture, which accounts for the spatial relationships of the living tissue. This way it is possible to more accurately reproduce the actual conditions of the body in a laboratory [30,31].

When planning an effective culture of cartilage cells, two main factors must be considered to ensure that the in-vitro culture is compatible with the in-vivo one. The first one is the relationships between the cell and its environment, i.e., both biochemical and physical aspects. Their effects on cellular response are investigated by mechanobiology [32,33]. An optimal culture environment maximally similar to in-vivo conditions in terms of temperature, access to the medium and other elements mentioned above, is necessary for the proper growth and development of the culture. The second important factor that should be analyzed when investigating the occurrence and development of osteoarthritis is the effect of mechanical stress on cartilage cells.

Each tissue is made up of cells that interact with each other and also react to external factors. Therefore, any force applied to the body causes reactions not only on the macroscopic scale, visible to the human eye, but also on the microscopic level as it affects the cells. The way the cells respond to the surrounding environment results directly from their mechanical and biochemical properties. It seems that the mechanical properties have a key impact on the strength parameters of tissues. Tissue engineering, the aim of which is to produce in a laboratory fully functional tissues that can replace damaged parts of the human body, has significantly contributed to the development of mechanobiology. In-vivo, cells receive signals from neighboring cells, react to information transmitted by enzymes, and adapt to changes in their environment. To recreate the mechanisms operating at the cellular level in laboratory conditions, it is necessary to investigate the above-mentioned factors and determine their impact on the final result. A change in cells’ functioning may entice a modification of their strength parameters and, consequently, of the biochemical signals they would send to the other cells. Therefore, to establish a culture that can be reliably used to investigate the functioning of the entire organ, it is necessary to carefully analyze the cells or cell aggregates and treat them not only as a biological but also a mechanical element. 

Bearing all these factors in mind, the researchers have been developing the aforementioned types of 2D and 3D cultures to provide cells with conditions as close as possible to in-vivo ones and also to make these studies as cost-effective as possible. 

### 3.1. Monolayer Cell Cultures

Adherent (2D) cultures are usually carried out on a flat surface, most often in a Petri dish, a multi-well polystyrene plate, or a culture flask. Polystyrene is a transparent plastic material, with the type of porosity that makes cell adhesion very effective. 

The greatest advantages of a 2D culture are its simplicity and high efficiency, and the possibility of producing a large culture from a small sample. Moreover, as the cells are arranged in a single layer, it is possible to provide all of them with the same amount of nutrients, which increases the probability of obtaining a stable and uniform level of cell growth and proliferation [34].

A disadvantage of this solution is its low similarity to in-vivo conditions. In a living organism, the cells receive and react to stimuli from all dimensions. The isolation of a single layer of cells from the spatial structure destroys specific natural interactions [35]. Moreover, most 2D cultures do not allow for free flow of the medium, which is yet another deviation from natural conditions, where nutrients freely reach the cells and the waste products are gradually removed [36,37,38]. 

Monolayer cultures of chondrocytes are quite effective in testing the effects of selected mechanical loads or the substrate. A group of US scientists investigated the effect of hydrostatic pressure on the biochemical reactions (factors such as aggrecan, collagen of type I and II, and β-actin) in chondrocytes [39]. The literature provides more examples of testing the load in monolayer cultures [40]; however, it should be remembered that to restore the proper mechanotransduction of chondrocytes, the extracellular matrix (ECM) is necessary, in which the cells freely perceive stimuli and transmit signals in space [41].

### 3.2. Spatial Cell Cultures

The second type of cell cultures are spatial (3D) cultures that allow for in-vitro reconstruction of a characteristic, spatial tissue system, including the ECM. Under these conditions the cells can interact with each other, respond to biochemical and biomechanical stimuli from the surrounding structures, or stimulate the expression of selected genes that ensure proper functioning of the tissues [42,43]. In a spatial culture the biochemical information is transferred between the cells and between the cells and the ECM in the same way as in the body [44]. 

Spatial cell cultures can have different forms. The main difference is establishing the culture with or without a scaffold.

#### 3.2.1. Scaffoldless

In a scaffoldless culture the cells are forced to clump together by shaking or vortexing the culture vessel. Chondrocytes form so-called aggregates and spheroids up to 5 mm in diameter that functionally resemble a tissue [45,46]. According to one definition, an aggregate is a macromolecular compound that is an essential component of the cartilage matrix, while spheroids are spatial clusters of cells with a spherical shape, comprising the aggregates. Thanks to their spatial structure, they allow for laboratory (in-vitro) analysis of the cell-cell and cell-ECM interactions. Importantly, the compressive strength of such a cartilage-like structure is over one third of that of the actual cartilage tissue [47]. This is important information when conducting mechanical research typical of tissue engineering focused on OA. Spheroids are characterized by a layered structure, with different properties of particular zones. The inner layer, known as the necrotic core, is formed as a result of hampered diffusion of the toxic waste outside the spheroid and low availability of oxygen and nutrients. The middle layer is the area of equilibrium between the nutrients supplied to the inside and the toxins removed outside. The outer zone of the spheroid is the area where cell proliferation is very intense. This is due to the unlimited access to oxygen and nutrients, and the ease of removing harmful metabolites. Spheroids are used for works analyzing cell metabolism, cell proliferation and differentiation, intercellular interactions, ECM synthesis, and vascularization of cell clusters. A disadvantage of this solution is short viability of cells due to the central necrosis resulting from impaired access of nutrients. Cartilage-like spheroids can also be obtained by growing cells in a hanging drop [48]. However, the scaffoldless cultures have a less organized structure than the cartilage tissue in a living organism [49].

#### 3.2.2. Scaffolds

In in-vivo cultures, the cells are properly arranged in space thanks to the ECM, which not only provides cells with an appropriate scaffold, but also regulates their reproduction and differentiation. To recreate an in-vivo culture in laboratory conditions, it is necessary to use a matrix substitute made of appropriate materials and fulfilling strictly defined functions. Scaffolds solve the problem of possible necrosis in the central part of the culture by providing a structure with parameters that allow free flow of the medium with nutrients or information in the form of proteins. The porous structure enables the nourishment of cells embedded in the inner pores of the scaffold (Figure 3) [1].

Scaffolds should be made of materials ensuring appropriate porosity and permeability. In addition, the material from which the culture scaffold is made must be at least biocompatible, that is must have appropriate surface roughness that allows for cell deposition on its surface. A proper scaffold should have mechanical properties close enough to those of the cartilage to allow the sample with cells to be subjected to loads similar to actual ones [50]. The loads to which cells respond when the tissue is not yet damaged are in the range of 6 to 20 MPa [51]. Depending on the material used to make the scaffolds, they can be divided into natural, synthetic, and composite ones.

Considering the above information, the most important functions of the cell culture scaffold are as follows:Creating a spatial support for the cultured cells,Ensuring proper adhesion of the cells to the substrate,Enabling development and proliferation of cells through appropriate geometric properties,Ensuring living tissue-like mechanical properties of the entire culture structure.

An additional advantage of the scaffold-based culture is the possibility of testing the cell response to various factors and controlling their response by incorporating specific elements into the scaffold structure. These can be for example bioactive, growth factor-rich layers coated on the surface of the scaffolds, as described in detail by Bjelić and Finšgar in 2021 [52]. 

In order for the scaffold to have the expected functions, it is also necessary to take into account additional features of the material from which the scaffold is built. The material should or can have the following properties:Biocompatibility–the material cannot be toxic and cannot cause an immune system reaction,Biodegradability–the scaffold material may degrade over time and under certain conditions,Specific surface properties–the scaffold material must have an appropriate structure ensuring cell adhesion to the substrate,Porosity–the material must ensure the possibility of growing a large number of cells with a relatively small volume of the substrate, and must allow for free migration of cells and even supply and removal of nutrients,Appropriate mechanical properties–the material should meet the specific requirements for a given type of culture.

Accordingly, the materials used to build the cell culture scaffold may be, as indicated above, of natural, synthetic, or composite origin.

Natural scaffolds are mostly based on collagen and hyaluronic acid (hyaluronan/hyaluronic acid), which are found in natural cartilage tissue. Due to their origin, these scaffolds are recognized by cartilage cells and significantly influence the course of the culture, for example, by supporting cell proliferation and building of the cartilage tissue. Collagen, as a protein found in the human body, including the intercellular fluid, skin, tendons, hair or nails, meets the requirements for biocompatibility.

Moreover, collagen is also a component of connective tissue and cell matrix [53]. However, it is necessary to remember the often insufficient strength properties of materials based on collagen or hyaluronic acid, and the fact that scaffolds made of natural compounds lose their strength and structural properties over time [54].

In order to obtain selected, optimal mechanical properties of the scaffolds, some studies focused on degenerative changes utilize scaffolds made of natural substances derived from outside the human body, such as alginates obtained from seaweed or silk characterized by a fibrous structure [55,56]. Materials based on silk may have diverse structural forms such as fibrous, porous or thin film. The properties such as their versatility, biodegradation, and biocompatibility as well as favorable capability of strengthening attachment, proliferation, and differentiation of chondrocytes are crucial to applying this kind of scaffold in tissue engineering [57,58].

Synthetic scaffolds allow for customization of physical and chemical properties during their manufacture. Thanks to their appropriate features and biodegradability, synthetic polymers are increasingly used in tissue engineering. The most popular synthetic polymers used for the production of culture scaffolds are polylactide (PLA), polyglycolide (PGA), and polycaprolactone (PCL). A very important property of synthetic polymers is their biodegradability, that is gradual degradation of the material into substances that are safe for the body, such as carbon dioxide and minerals. The time of biodegradation is controlled and depends on the type of culture and, inter alia, on the diameter of the fibers. Current technology allows for programming selected material or biochemical properties depending on the chemical composition of the raw material. An alternative to collagen is an elastin-like polymer. Elastin is present in the human body as a component of connective tissue, intercellular matrix, tendons, ligaments, skin, and walls of large blood vessels. The scaffolds made of elastin-like polypeptides feature properties similar to those of elastin, primarily flexibility [59]. These properties are also helpful in research involving the cultures of fibroblasts, the skin-building cells [50].

An interesting solution in this field is the use of composite scaffolds. Design of the composite material for a scaffold for cell culture requires the appropriate selection of materials for the matrix and for the modifying phase, with biocompatibility being the basic, decisive criterion. Modifying phases can have the form of a powder, porous structures and solid, dense materials, while the matrix is made, for example, from fibers [60] (Figure 4).

Tissue engineering applied in studies on osteoarthritis-related factors and cartilage repair increasingly use scaffolds printed on bioprinters, where cells are seeded onto consecutive layers of the print [61,62].

To make the culture scaffolds as similar as possible to the structure of cartilage, they can be produced as hydrogels, nanofibers, foam or sponge [1]. These structures are porous and permeable, but the production of an uncontrolled structure can be a limitation. The technology of bioprinting may provide a solution to the problem of the lack of control over the structural properties, as it enables planning the geometry of the hydrogel structure and simultaneous seeding of cells [63]. The hydrogels used in this technology can be made of both natural and synthetic polymers. By selecting polymers and combining them in various proportions, it is possible to obtain hydrogels with different mechanical properties, tailored for a given research task [64].

A cell culture, based on a hydrogel structure, begins with combining the cells with a liquid pre-hydrogel, which is then exposed to external factors such as, for example, radiation of a given wavelength, temperature, or appropriate chemicals. When exposed to the selected factor, the polymer begins to gel and swell [65], resulting in the formation of the expected structure intended for the cell culture.

#### 3.2.3. Explants

Research in an isolated tissue fragment called an explant, is another solution employed in tissue engineering [37]. When conducting research in cell cultures, it is important to ensure conditions that guarantee the development of the research material imitating the investigated organ. This material can perform the same functions as in-vivo tissue. 

The simplest solution is to isolate a fragment of a living tissue. This allows for reducing costs and simplifying the research material production, and provides all the structural conditions and mechanical properties of a culture so important in OA-oriented research [66]. Unfortunately, obtaining explants can be very difficult due to the procedures and availability of explants derived from living organisms. The advantage of explants is not only the fact that they do not grow, but also that they maintain the functionality of cells which, reacting to external stimuli, are consequently capable, for example, of releasing specific biochemical factors. A disadvantage of such a culture is its short viability (up to 10 days) [56].

#### 3.2.4. Decellularized Composite Scaffolds

A very interesting type of scaffolds are hybrid scaffolds which incorporate decellularized matrix with synthetic biomaterials. Generally, decellularization is utilized to remove cellular components from articular cartilage. It is needed to preserve structural proteins and other molecules during the process of decellularization. Gel created from this decellularized ECM is mechanically more similar to the native cartilage tissue and capable of inducing chondrogenesis. Once a decellularized ECM is prepared, it can be coated onto culture dish or multi-hole plates as well as onto a scaffold prepared for instance by 3D prototype printing techniques. Application of this type of scaffold is very promising in regeneration tissue study [67,68,69,70].

## 4. Bioreactors

In in-vivo models, nutrients and body fluids are delivered to the cells continuously rather than intermittently. Therefore, even many three-dimensional models of cell cultures, e.g., spheroids, in which the medium is administered to the cells from time to time, do not fully reflect the natural conditions. In-vitro cultures that enable continuity of physiological processes and provide control over the supplied compounds can be grown in bioreactors. These complex systems are usually used for spatial cultures but, depending on the needs, monolayer cell cultures can also be grown this way [71]. Due to their complexity, these systems are often called bioreactors.

Cell cultures require constant monitoring of numerous parameters, i.e., temperature, pH, access to nutrients and oxygen, as well as other physical and chemical parameters depending on the purpose of the culture. A bioreactor is a device that facilitates the control of all factors influencing the final effect. There are many literature definitions of bioreactors: “A bioreactor is the general term applied to a closed culture environment, that is usually mixed, that enables control of one or more environmental or operating variables that affect biological processes” [72], “The role of a bioreactor is to mimic the in-vivo conditions of a tissue. The bioreactor should be able to control the different variables that define the environment of the tissue” [73], “Bioreactors are devices in which biological or biochemical processes develop under a closely monitored and tightly controlled environment” [74]. These definitions clearly show that bioreactors used in tissue engineering can be defined as systems that enable cell cultures to be carried out in conditions similar to natural ones and in which the monitoring of variable biomechanical parameters is possible. Depending on the type of culture, the bioreactors can have various structural and technical parameters. 

The main advantage of bioreactor cultures is the possibility of providing the cells with specific substances or other factors, and then monitoring their effect on the cell culture. In the case of cartilage engineering, mechanical loads are an important factor influencing the development of the cultures. This is due to the fact that in a living organism, chondrocytes are subjected to steady and cyclical loads caused by body movements. Therefore, in order to best reproduce in-vivo conditions in the laboratory, it is worth using a bioreactor that enables not only the introduction of additional loads, but also controlling their impact on the cells [75]. 

Tissue engineering was used in research on the possibility of in-vitro production of a cartilage-like structure that meets the mechanical requirements for a natural tissue and its implantation in a patient. Implementation of this purpose required three key elements: autologous cartilage cells (chondrocytes), a scaffold on which the cells can be placed, and a bioreactor, i.e., a device reproducing in-vivo culture conditions. 

### Types of Bioreactors for Chondrocyte Culture

Depending on the type of load to which the culture is subjected, the literature distinguishes several basic designs of bioreactors. Considering how joints, and specifically, cartilage tissue, work and how mechanical loads affect the development of osteoarthritis, apart from controlling biophysical factors, the most important aspects seem to be shear force and compression. Shear load bioreactors can implement the load by shear contact [76] or fluid flow [77], for example, in so-called rotating wall vessel (RWV) bioreactors [78]. Pressure bioreactors, on the other hand, can be divided into indirect pressure systems (Figure 5), in which hydrostatic pressure is generated [79], and direct pressure systems using various types of pistons [80].

Indirect pressure bioreactors have a piston-like element that creates hydrostatic pressure by compressing the gas that transfers the load to the medium surrounding the cell culture or by squeezing the medium. In direct pressure bioreactors the piston compresses the culture fluid directly, bypassing the gaseous layer [81].

Shear bioreactors provide a friction load as a result of contact shear force between the working surface and the culture structure or hydrodynamic shear in the form of the fluid flow shear [82]. The first type of direct shear bioreactors is equipped with a surface moving over the culture structure [83]. The second type of bioreactors are, for example, RWV ones, in which the outer cylinder making up the bioreactor chamber rotates and causes the fluid to vortex. Spinner Flask systems provide another example, in which the fluid movement is triggered by a rotating bar placed at the bottom of the chamber [84]. A solution that can be used to generate friction load is perfusion of a spatial, porous culture structure in so-called perfusion systems. Fluid flow control is often enforced by peristaltic pumps that ensure a linear flow velocity of the medium [85,86].

The RWV-type bioreactor is considered the best device for the culture of cartilage cells due to the optimized method of delivering nutrients [87]. Various variants of this construction are employed, for example rotation-based bioreactor systems [88]. The rotating part is the base of the cylinder, and the scaffolds can be freely suspended in the medium or attached to the curved walls of the bioreactor. The culture can also be seeded on cylindrical elements attached to the central shaft connected to the rotating base of the cylinder (Figure 6). 

Research results published in 1996 demonstrated that the Spinner Flask bioreactor (see Figure 7) allowed for obtaining chondrocytes of a very regular shape, and the number of proliferated cells was up to 70% greater than in static cultures [89]. 

Direct friction bioreactors are equipped with an element whose surface moves on the surface of the cell culture with a light axial pressure. This movement can cause sliding friction or rolling resistance as shown in the exemplary design of the structure in Figure 8 [76,90]. The friction comes from the ceramic hip ball that makes cyclic movements compressing the scaffold with simultaneous rotation, which causes friction on the culture surface (combination of compression and shear forces). 

Flow perfusion bioreactors are equipped with a pump that forces the flow of the medium through the culture structure and controls its velocity. In addition, the cells must be seeded on porous and permeable scaffolds to allow the medium to flow inside the structure. Thanks to this and the ability to control selected factors, including the flow velocity of the medium, the cells are provided with a continuous supply of nutrients with simultaneous tangential loading, as in natural cartilage. 

A combination of a flow perfusion bioreactor and a scaffold of a porous structure make it possible for the medium to reach all cells of the culture. An example of this system is a design proposed by German researchers in 2012 [91] and presented in Figure 9. With appropriate regulation of the fluid flow velocity, it is possible to obtain a tissue with a homogeneous structure and biomechanical properties corresponding to natural cartilage [92].

## 5. Lab-on-Chip Systems

Another solution bringing the in-vitro systems closer to the in-vivo ones is the use of the modern method of flow microsystems, also known as lab-on-chip (LoC). It is a microlaboratory system, built of many properly arranged elements and forming an in-vitro microplatform resembling an in-vivo culture. The LoC microsystems are equipped with cell culture sites, microchannels supplying the cells and a whole network of microstructures, through which it is possible to constantly supply the medium and control selected biomechanical factors, as in Figure 10 [93].

As in bioreactors, cell cultures in lab-on-chip systems are provided with a continuous flow of the medium through a flat or three-dimensional culture structure. Additionally, the lab-on-chips enables studies on the interaction not only between different types of tissues but also between individual organs [95,96,97]. Moreover, these complex systems go beyond 2D culture research. Depending on the design of the chip, it is also possible to conduct experiments on three-dimensional cultures produced, for example, by bioprinting [98]. The flow platforms make it also possible to run co-cultures, and thus enable the analysis of the interactions of various cell lines forming tissues, organs, or whole organisms [94]. An example of such an extensive lab-on-chip system, taking into account cell cultures present in various joint tissues, is presented in [99].

The above example of a joint-on-chip platform illustrates just how complex and multi-layered a lab-on-chip architecture can be. However, it should be remembered that the more complex the culture systems are, the more difficult they are to control, and the more expensive it is to produce such carriers.

## 6. Conclusions

The aim of tissue engineering is to search for solutions for conducting cell cultures that enable to study the processes taking place at the cellular level, to be able to affect them in a controlled way, and finally to propose new treatment methods. Depending on the structural complexity of the culture carriers and the investigated problem, research can be carried out in simple, monolayer, individual cell cultures or more complex systems of 3D cultures. It seems that, due to the etiology of the degenerative disease, research on the processes accompanying OA should be aimed at lab-on-chip systems; however, when expanding the culture systems one should always bear in mind that the greater their complexity, the higher the costs of their production and of conducting the research. Nevertheless, it can be assumed that the cell culture platforms used for OA research will be developed primarily in the field of material technologies and structures enabling co-cultures. The need for new methods of treating OA forces the development of tissue engineering and modern culture scaffolds enabling the possible implantation of new tissue [100]. In addition to innovative culture biomaterials, an interesting field of research is also designing new structures of bioreactors and flow microsystems that would enable the analysis of individual stages of osteoarthritis development.

## Figures and Tables

**Figure 1 ijms-23-10308-f001:**
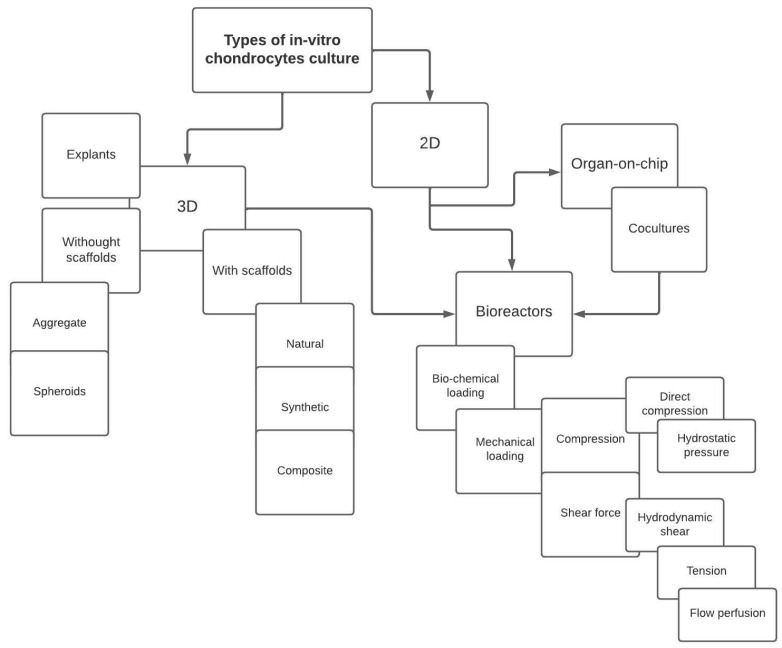
Diagram showing solutions for in-vitro cultivation of cartilage cells.

**Figure 2 ijms-23-10308-f002:**
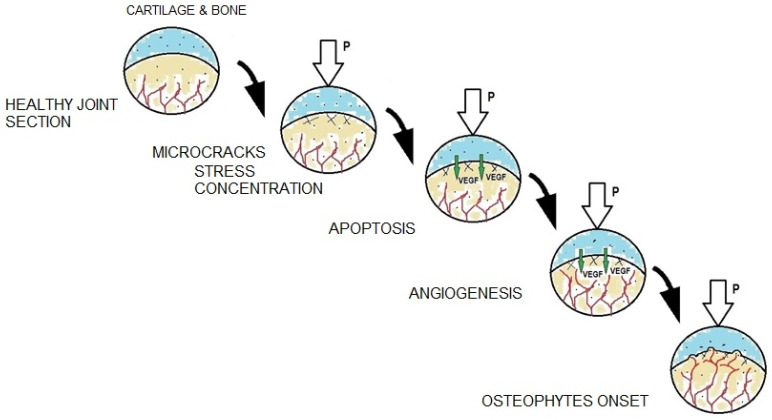
Mechanical and biological processes accompanying individual stages of degenerative changes.

**Figure 3 ijms-23-10308-f003:**
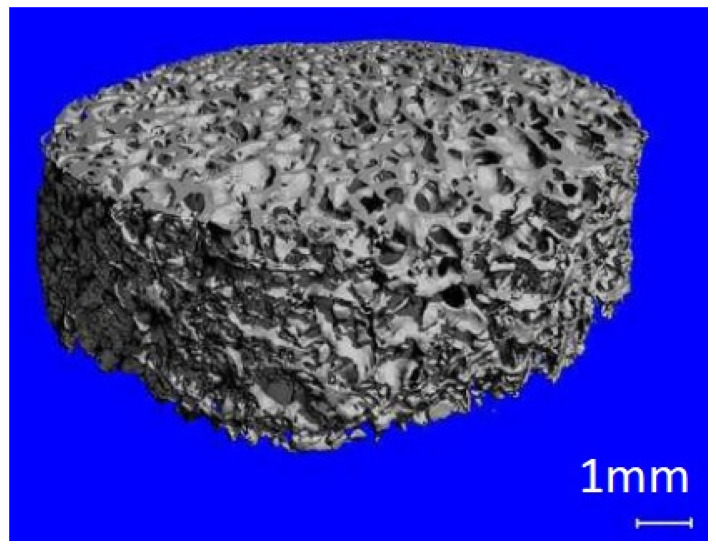
Decellularized cancellous bone scaffold.

**Figure 4 ijms-23-10308-f004:**
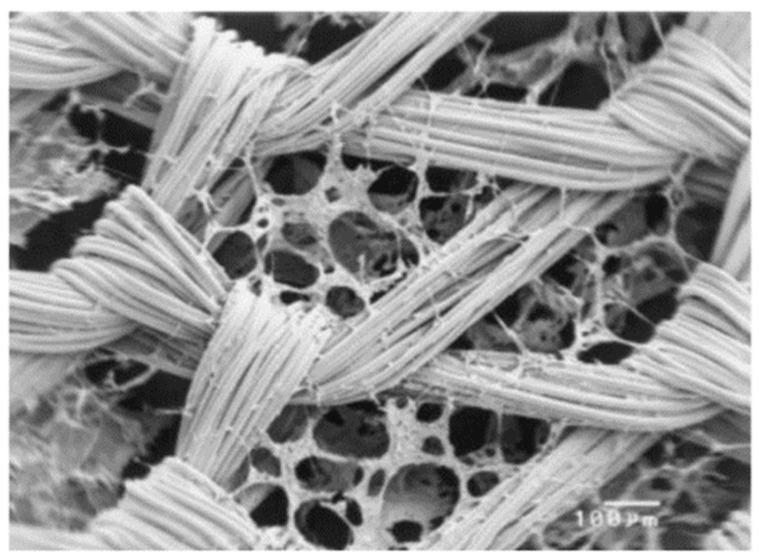
SEM photomicrograph of knitted non-woven polyglactin (PLGA) fiber/collagen composite scaffold (scale bar is 100 µm). Reprinted with permission from [60]. Copyright 2003, Wiley Periodicals, Inc., Hoboken, NJ, USA.

**Figure 5 ijms-23-10308-f005:**
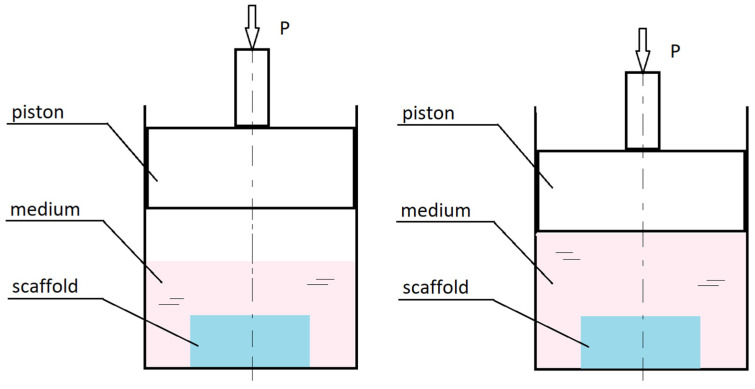
Diagram showing how intermediate pressure loads work in culture bioreactors.

**Figure 6 ijms-23-10308-f006:**
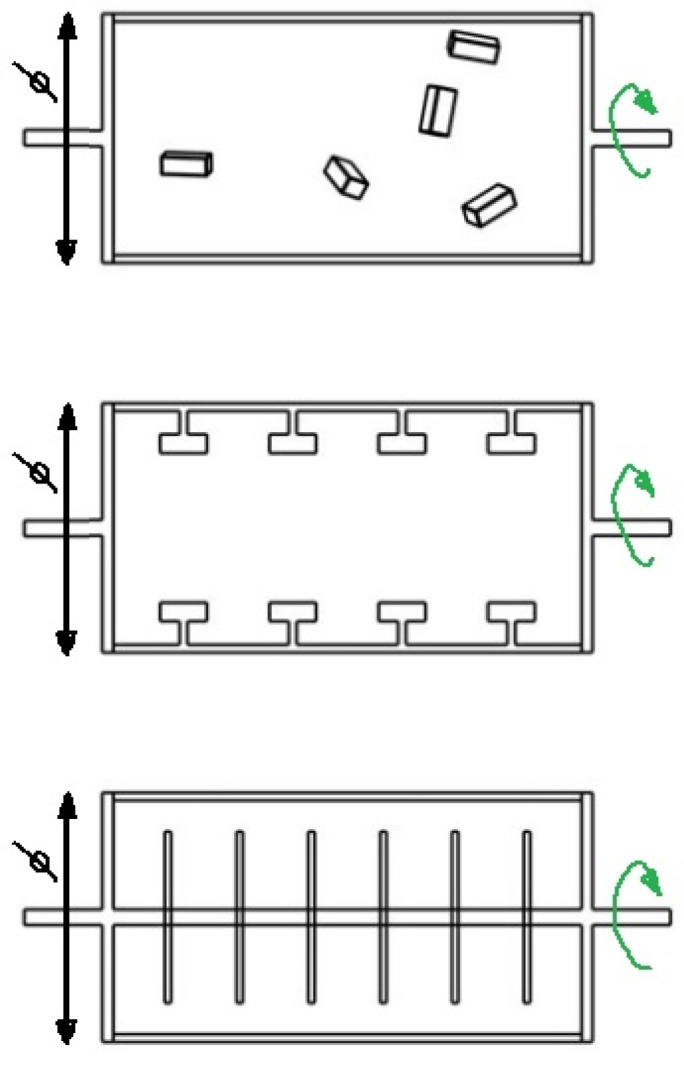
Types of rotation-based bioreactor systems.

**Figure 7 ijms-23-10308-f007:**
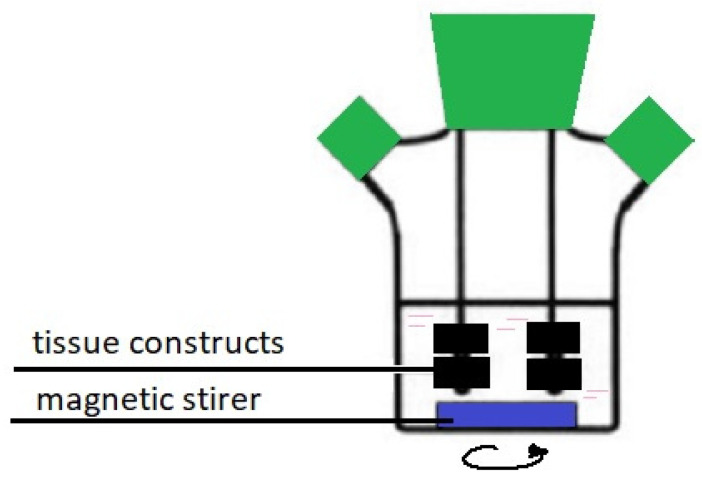
Spinner Flask bioreactor.

**Figure 8 ijms-23-10308-f008:**
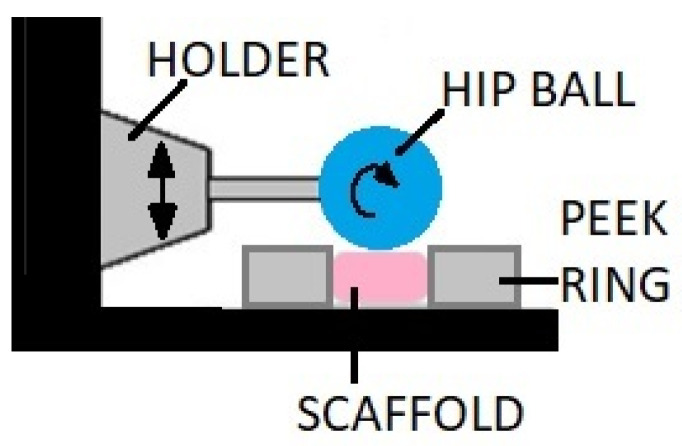
A bioreactor with mechanical loading provided by the ceramic hip ball.

**Figure 9 ijms-23-10308-f009:**
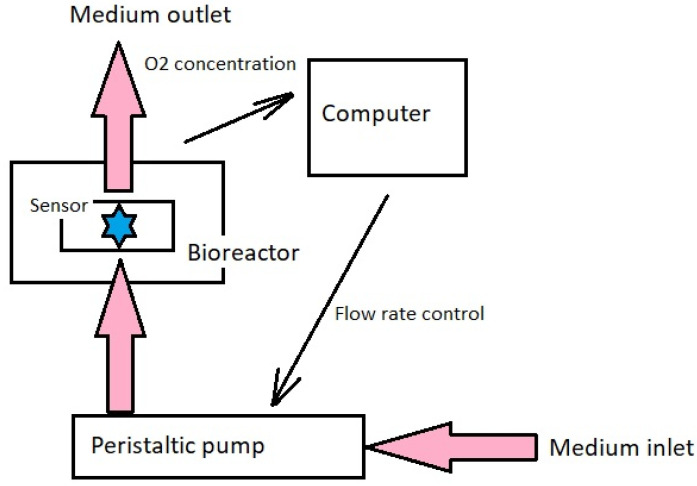
System providing medium flow in a culture bioreactor.

**Figure 10 ijms-23-10308-f010:**
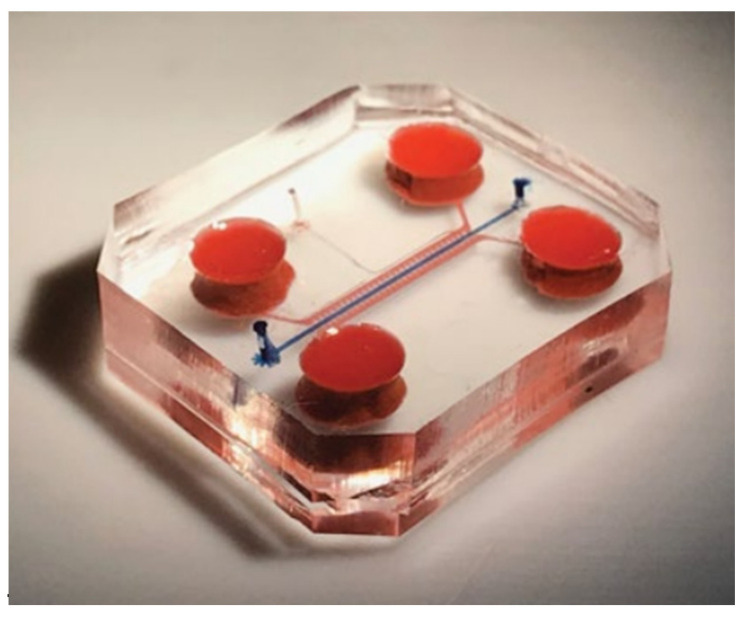
An example of a lab-on-chip system consisting of a lower and upper body part, medium supply channels and a culture channel. Reprinted with permission from [94]. Copyright 2019, Nature Biomedical Engineering.

**Table 1 ijms-23-10308-t001:** Causes and effects of osteoarthritis.

Cause	Symptom/Effect
Previous injuries	Reduction of the cushioning properties of cartilage
Obesity	Reduced amount of synovial fluid
Age	Pain and inflammation in the joint area
Sex	Microstructural changes in cartilage tissue
Mechanical overload	Cartilage fibrosis
Genetic predispositions	Initiation of angiogenesis
Cartilage wear	Growth of osteophytes

## Data Availability

Not applicable.

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
