# Peer review of "Chondrocytes In Vitro Systems Allowing Study of OA"

_ijms, 2022, doi:10.3390/ijms231810308_

Round 1
Reviewer 1 Report
The manuscript deals with highlighting different chondrocytes in-vitro systems along with the use of diverse biomaterials for application in cartilage regeneration aiming to treat Osteoarthritis. The manuscript needs some revision, below are the few issues
1) Figure 3: Please include the scale bar for this figure.
2) Line 324 to 330: Authors are requested to rewrite the following lines, since its matching with the existing literature online.
3) Authors are requested to add a separate section titled “Decellularized composite scaffolds” to include all the literatures detailing hybrid scaffolds made by incorporating decellularized matrix with synthetic biomaterials
4) Authors are requested to incorporate literatures from decellularized human placental tissue for cartilage regeneration
5) Figure 4: Scales bar is hardly visible in the SEM photomicrograph
6) Authors are requested to include more literatures highlighting “silk for cartilage tissue engineering” since silk has been demonstrated as an attractive/versatile material for cartilage regeneration
Author Response
Response to Reviewer 1 Comments
Dear Reviewer,
I would like to thank you for reading and providing comments on my manuscript. I am very appreciated for your constructive criticism which helped me improve the text. Please find below the responds and clarifications to all of the comments. The tracked changes version has been submitted in addition to a “ijms-1872946” version of the document.
Point 1. Figure 3: Please include the scale bar for this figure.
Response 1: Thank you for the comment, Figure 1 scale bar has now been included.
Point 2. Line 324 to 330: Authors are requested to rewrite the following lines, since its matching with the existing literature online.
Response 2: Thank you for your comment. In the lines indicated I have cited and compared existing definitions so I would prefer not to change the text as it was planned out to expose the main characteristic of bioreactors.
Point 3. Authors are requested to add a separate section titled “Decellularized composite scaffolds” to include all the literatures detailing hybrid scaffolds made by incorporating decellularized matrix with synthetic biomaterials
Response 3: Thank you for the comment. I have revised the text and included the following paragraph:
“Decellularized composite scaffolds A very interesting type of scaffolds are hybrid scaffolds which incorporate decellularized matrix with synthetic biomaterials. Generally, decellularization is utilized to remove cellular components from articular cartilage. It is needed to preserve structural proteins and other molecules during the process of decellularization. Gel created from this decellularized ECM is mechanically more similar to the native cartilage tissue and capable of inducing chondrogenesis additionally. Once a decellularized ECM is prepared, it can be coated onto culture dish or multi-hole plates as well as onto a scaffold prepared for instance by 3D prototype printing technic. Application of this type of scaffold is very promising in regeneration tissue study [68, 69, 70, 71].”
Point 4. Authors are requested to incorporate literatures from decellularized human placental tissue for cartilage regeneration
Response 4: Thank you for your request. I have added literatures detailing hybrid scaffolds made by incorporating decellularized matrix used for tissue engineering:
- Rijal G. The decellularized extracellular matrix in regenerative medicine. Regenerative Medicine. 2017; 12(5):475–477. doi:10.2217/rme-2017-0046.
- Ghosh P, Gruber SMS, Lin C-Y, Whitlock PW. Microspheres containing decellularized cartilage induce chondrogenesis in vitro and remain functional after incorporation within a poly(caprolactone) filament useful for fabricating a 3D scaffold. Biofabrication. 2018; 10(2):025007. doi:10.1088/1758-5090/aaa637.
- Beck EC, Barragan M, Tadros MH, Gehrke SH, Detamore MS. Approaching the compressive modulus of articular cartilage with a decellularized cartilage-based hydrogel. Acta Biomaterialia. 2016; 38:94–105. doi:10.1016/j.actbio.2016.04.019.
- Sutherland AJ, Detamore MS. Bioactive Microsphere-Based Scaffolds Containing Decellularized Cartilage. Macromolecular Bioscience. 2015; 15(7):979–989. doi:10.1002/mabi.201400472.
Point 5. Figure 4: Scales bar is hardly visible in the SEM photomicrograph
Response 5: Thank you for the comment. The figure was expanded to better show the scale bar.
Point 6. Authors are requested to include more literatures highlighting “silk for cartilage tissue engineering” since silk has been demonstrated as an attractive/versatile material for cartilage regeneration
Response 6: Thank you for your comment. I have revised the text and included the following paragraph with more literature:
“Materials based on silk may have a diverse structural forms such as fibrous, porous or thin film. The properties such as their versatility, biodegradation, and biocompatibility as well as favorable capability of strengthening attachment, proliferation, and differentiation of chondrocytes are crucial to applying this kind of scaffolds in tissue engineering [58, 59].”
- Chao P-HG, Yodmuang S, Wang X, Sun L, Kaplan DL, Vunjak-Novakovic G. Silk hydrogel for cartilage tissue engineering. Journal of Biomedical Materials Research Part B: Applied Biomaterials. 2010; 95B(1):84–90. doi:10.1002/jbm.b.31686.
- Cheng G, Davoudi Z, Xing X, Yu X, Cheng X, Li Z, et al. Advanced Silk Fibroin Biomaterials for Cartilage Regeneration. ACS Biomaterials Science & Engineering. 2018; 4(8):2704–2715. doi:10.1021/acsbiomaterials.8b00150.
Yours sincerely,
Ewa Bednarczyk
Reviewer 2 Report
The manuscript titled "Chondrocytes in-vitro systems allowing study OA" is a comprehensive literature review comparing the different cultivation methods and options to better understand OA.
A few minor things should be corrected before it is published.
- Line 216: where are REF 53 and 54? REF 52 is on line 216 and
then it suddenly continues with REF 55 on line 229
- REF 60 is not appropriate because it talks about Development of Intra-Articular Drug Delivery Systems and not elastin
- and where is REF 61 in the body text? REF 60 was still on line 274 and then it continues with REF 62 in line 282
- REF 72 is on line 343 and then it continues with REF 75 on line 354 .....
all the best and with kind regards
Author Response
Response to Reviewer 2 Comments
Dear Reviewer,
I would like to thank you for reading and providing comments on my manuscript. I am very appreciated for your constructive criticism which helped me improve the text. Please find below the responds and clarifications to all of the comments. The tracked changes version has been submitted in addition to a “ijms-1872946” version of the document.
Point 1. Line 216: where are REF 53 and 54? REF 52 is on line 216 and
then it suddenly continues with REF 55 on line 229
Response 1: Thank you for the comment. REF 53 and 54 were repeated in other REF. I have revised the references and fixed the numbering.
Point 2. REF 60 is not appropriate because it talks about Development of Intra-Articular Drug Delivery Systems and not elastin
Response 2: Thank you for your comment, I have changed the reference to:
Chen Z, Zhang Q, Li H, Wei Q, Zhao X, Chen F. Elastin-like polypeptide modified silk fibroin porous scaffold promotes osteochondral repair. Bioactive Materials. 2021; 6(3):589–601. doi:10.1016/j.bioactmat.2020.09.003.
Point 3. and where is REF 61 in the body text? REF 60 was still on line 274 and then it continues with REF 62 in line 282
Response 3: Thank you for pointing out. Missing REF 61 was only in the Figure 4 caption so I have added it in the text.
Point 4. REF 72 is on line 343 and then it continues with REF 75 on line 354 ....
Response 4: Thank you for your comment, I have revised and reordered the references.
Yours sincerely,
Ewa Bednarczyk
Round 2
Reviewer 1 Report
Manuscript can be accepted.
Reviewer 2 Report
The author has carefully revised all suggestions for improvement, so that the work can now be published.
Congratulations
With kind regards